# The Role of lncRNAs in Pig Muscle in Response to Cold Exposure

**DOI:** 10.3390/genes14101901

**Published:** 2023-09-30

**Authors:** Dongjie Zhang, Liang Wang, Wentao Wang, Di Liu

**Affiliations:** Institute of Animal Husbandry, Heilongjiang Academy of Agricultural Sciences, Harbin 150086, China; djzhang8109@163.com (D.Z.); wlwl448@163.com (L.W.); wangwentao_1981@163.com (W.W.)

**Keywords:** pig, skeletal muscle, low temperature, lncRNA, functional analysis

## Abstract

Cold exposure is an essential factor affecting breeding efforts in cold regions. Muscle, as an important tissue for homeothermic animals, can produce heat through shivering thermogenesis (ST) and non-shivering thermogenesis (NST) under cold exposure. Long non-coding RNAs (lncRNAs) play important roles in regulating gene expression. However, the regulatory mechanisms of lncRNAs and their role in the thermogenesis of pigs are unclear. We examined lncRNAs in the skeletal muscle of an indigenous pig breed, the Enshi black pig, when the pigs were exposed to acute or chronic cold. Three pigs were maintained inside a pig house (control group), three pigs were maintained outside the pig house for 55 d (chronic cold group), and three pigs were suddenly exposed to the conditions outside the pig house for 3 days (acute cold group). After the experiment, the longissimus dorsi of each pig were collected, and their lncRNA profiles were sequenced and analyzed. Each sample obtained nearly 12.56 Gb of clean data. A total of 11,605 non-coding RNAs were obtained, including 10,802 novel lncRNAs. The number of differentially expressed lncRNAs (DElncRNAs) was identified under acute cold (427) and cold acclimation (376), with 215 and 192 upregulated lncRNAs, respectively. However, only 113 lncRNAs were commonly upregulated by acute cold and cold acclimation. In addition, 65% of the target genes were trans-regulated by DElncRNAs. The target genes were enriched in signal transduction, immune system, cell growth and death pathways, and amino acid and carbohydrate metabolism. Compared to cold acclimation, acute cold stress-induced more DElncRNAs and response pathways. In conclusion, low temperatures altered the expression levels of lncRNAs and their target genes in muscle tissue. Some potential mechanisms were revealed, including ion migration and the metabolism of amino acids and carbohydrates.

## 1. Introduction

Pigs were domesticated approximately 10,000 years ago, and at present, they are widely distributed, including in the polar regions and high plateaus. Early domesticated pigs possessed a high degree of climatic adaptability. However, with the rise of intensive breeding modes geared to maximize economic benefits, pigs have been provided with the most suitable growth environments [1], resulting in increasingly poor adaptability to environmental changes. Newborn piglets and pigs raised in cold or hot areas require heating or cooling. Although intensive breeding has led to increased sensitivity of pigs to environmental variation, there are indigenous breeds that retain their tolerance to extreme climatic conditions. These include Tibetan wild boars, a local breed living in the Qinghai-Tibet Plateau at an average altitude of 4268 m. This breed can tolerate both low temperatures and hypoxic conditions [2]. The Mangalica pig, the only pig breed with long hair, lives in the colder regions of Hungary and Romania, which is colder than Hungary [3]. The Tswana pig is a hardy breed that is well-adapted to the harsh climatic conditions of Botswana [4]. However, compared to small mammals that hibernate and cold-adapted insects [5], the mechanisms of cold adaptation in pigs are not well characterized due to their large size and the difficulty in creating an artificial low-temperature environment.

It is well known that white adipose tissue (WAT) can be induced to turn into beige fat under cold conditions [6]. Under these conditions, the lipid composition and transcriptional processes of WAT are significantly altered. Beige and brown adipocytes are rich in mitochondria and can consume energy in the form of heat.

Skeletal muscle is another important thermogenic tissue. It has the capacity to induce energy-consuming futile cycles for both shivering and non-shivering thermogenesis (NST) [7]. In addition, nervous system regulation [8] and gut microbiota [9] are important components for maintaining temperature stability. Although pigs have a relatively large amount of adipose tissue, mutations in the *UCP1* gene prevent them from forming traditional brown adipose tissue (BAT). Only some beige fat is found in piglets [10]. As in pigs, BAT is absent in animal groups such as birds, monotremes, marsupials, and many eutherians [11]. The heat production mechanism of birds is relatively well understood; skeletal muscle is the main thermogenic tissue.

One mechanism of skeletal muscle heat production is based on Ca^2+^-slippage by sarcoplasmic reticulum Ca^2+^-ATPase (SERCA) and is controlled by the protein sarcolipin (SLN) [12]. Another mechanism is through the regulated uncoupling of oxidative phosphorylation in muscle mitochondria [13]. Skeletal muscle-based NST plays an important role in the thermoregulation of both birds and mammals [14].

Cold exposure in homeothermic animals normally causes acute and chronic physiological responses. Acute cold exposure leads to cutaneous vasoconstriction and shivering thermogenesis. These physiological responses decrease heat loss and increase metabolic heat production. Chronic cold exposure evokes NST adaptations in BAT and skeletal muscle, helping to maintain core body temperature and energy homeostasis at low temperatures [15]. The Enshi black pig is an important Chinese indigenous breed. It is known for its excellent production traits such as good meat quality, strong adaptability, and tolerance to roughage feeding [16]. It also has a wide range of adaptability to ambient temperatures. The breed is farmed in Hubei province, which has a subtropical monsoon climate, as well as in Heilongjiang province, which has a cold temperate continental monsoon climate (Supplement Appendix A). The breed is thus a good model for examining the mechanism of pig adaptation to cold conditions.

In this study, Enshi black pigs were used as research subjects. The pigs were divided into a chronic low-temperature acclimation group and an acute short-cold stress group. Using the RNA-seq method, the lncRNAs in the skeletal muscles of the pigs were then compared, and the differentially expressed lncRNAs between the two groups were screened and analyzed. The results of this study will provide a theoretical foundation for breeding cold-tolerant pig breeds in the future.

## 2. Materials and Methods

### 2.1. Experimental Design and Sample Collection

Enshi black pigs were provided by Huazhong Agricultural University. In October 2022, nine three-month-old female Enshi black pigs with similar body weights were randomly divided into three groups labeled A, B, and C. Each group included three individuals. The pigs were raised inside a pig house where the temperature was controlled at 18 ± 2 °C. When the experiment began, the individuals of group B were released and maintained outside. At that time, the highest temperature/lowest outdoor temperature was 12 °C/1 °C. After 55 days, the ambient temperature dropped to −17 °C/−24 °C. At that time, group A with three individuals was moved outside. After 3 days, the experiment was terminated. Group C with three individuals remained in the warm pig house during the experiment (Figure 1). At the end of the experiment, the nine pigs were slaughtered, and 100 mg of the dorsal muscle was taken from each pig and stored in liquid nitrogen.

### 2.2. cDNA Library Construction and Sequencing

RNA extraction and quality detection were performed by referring to Zhang et al. [17]. When constructing the cDNA library, ribosomal RNA was first removed from the total RNA, and then, the RNA was broken into short fragments of 250–300 bp. The fragmented RNA was used as a template; random oligonucleotides were used as primers to synthesize the first strand of cDNA, after which dNTPs were used as raw materials to synthesize the second strand of cDNA. The purified double-stranded cDNA was end-repaired, A-tailed, and ligated with sequencing adapters. AMPure XP beads were used to screen cDNAs of about 350–400 bp, and the second strand of U-containing cDNA was degraded using the USER enzyme. Finally, PCR amplification was performed to obtain a cDNA library. An Illumina PE150 platform (San Diego, CA, USA) was used to perform sequencing. Clean reads for subsequent analysis were obtained after raw data filtering, sequencing error rate checking, and GC content distribution checking. High-throughput sequencing was performed by Novogene Technologies Co., Ltd. (Beijing, China).

### 2.3. Identification of lncRNAs

Stringtie was used to splice the reads aligned to the genome into transcripts. The transcripts obtained by splicing each sample were merged using Cuffmerge software. Low-confidence single-exon transcripts were filtered from the transcript splicing results, and transcripts with the exon number so2 were selected. From these, transcripts with a length greater than 200 nt were then selected. Cuffcompare software was used to screen out the exon regions that were annotated in the database. Overlapping transcripts and lncRNAs in the database that overlapped with the exon region of the spliced transcripts were included in the subsequent analysis as database-annotated lncRNAs. For the transcripts obtained in the previous step, current mainstream coding potential analysis methods (CPC2/Pfam/CNCI) were employed to predict the coding potential, and the transcripts without coding potential were taken from the results as a candidate novel lncRNA dataset.

### 2.4. Analysis of DElncRNAs

The expression levels of lncRNAs were estimated by fragments per kilobase of transcript per million fragments mapped (FPKM). Using edgeR software for significant differential expression analysis, the screening criteria were |log_2_FC| > 2 and *p* < 0.05. The target genes of lncRNAs were predicted by co-location and co-expression analyses between lncRNAs and protein-coding genes. For the co-location regulation mechanism, the threshold of co-location was set to 100 kb upstream and downstream of a lncRNA. The co-expression analysis was carried out by examining the correlation in expression across multiple samples. The screening condition was a correlation coefficient greater than 0.95. Then, functional enrichment analysis (GO/KEGG) was performed on the target genes of the DElncRNAs, and the results were used to predict the main functions of the lncRNAs.

To investigate whether genes associated with GO (Gene Ontology, http://geneontology.org/ (accessed on 12 March 2023)) terms were differentially expressed, a hypergeometric *p*-value was calculated and adjusted into a q-value, with the background set to be the genes from the entire genome. GO terms with a q-value < 0.05 were considered to be significantly enriched, and GO enrichment analysis showed what biological functions the DEGs performed. KEGG (Kyoto Encyclopedia of Genes and Genomes, http://www.kegg.jp/) is a database that contains a collection of manually drawn pathway maps representing our knowledge of molecular interaction and reaction networks. Using the same method used for GO enrichment analysis, significantly enriched KEGG pathways were also identified.

### 2.5. lncRNA-mRNA Network Construction

The Cytoscape program was used to construct a DElncRNA-mRNA interaction network. The networks of the top 10 upregulated DElncRNAs of groups A and B and their target mRNAs were constructed separately.

## 3. Results

### 3.1. Identification of lncRNAs in the Dorsal Muscle of Pigs

After the quality control of each sample, the amount of clean data obtained was nearly 12.56 Gb, accounting for approximately 97.66% of the total. Analyses of the assembled transcripts and the lncRNAs were based on these clean data. Sequencing data filtering and the classification of raw reads are shown in Figure 2a. The mapping regions of clean reads are shown in Figure 2b. After the coding potential prediction of the transcripts, a total of 11,605 noncoding RNAs were obtained (Figure 2c), including 474 known lncRNAs and 10,802 novel lncRNAs. Among the novel lncRNAs, 42.14% were long intergenic non-coding RNAs (lincRNAs), with 37.47% of them being sense overlapping and 20.39% being antisense. The distribution of novel lncRNAs on each chromosome is shown in Figure 2d. Except for the Y chromosome, novel lncRNAs were relatively evenly distributed on the chromosomes.

### 3.2. Characterization of lncRNAs in the Dorsal Muscle

The characteristics of the acquired novel lncRNAs in Enshi black pigs’ dorsal muscles were examined. The novel lncRNAs with two exons were the most common (65.64%), followed by those with three exons (18.45%) (Figure 3A). The lengths of novel lncRNAs and the lengths of the open reading frames (ORFs) of novel lncRNAs were consistent with those of the annotated lncRNAs. The distribution of the length of novel lncRNAs was centered at 200–1000 bp (Figure 3B), and the of lncRNAs in this length range was 4982. Among the novel lncRNAs, 89.12% contained a short ORF (approximately 30–200 amino acids) (Figure 3C). There were significant differences between lncRNAs and mRNAs. After calculating the expression value (FPKM) of all lncRNAs in each sample, the distribution of the expression levels of lncRNAs in different samples was displayed in the form of box plots (Figure 3D).

### 3.3. Differential Expression (DE) of Cold Stress-Related lncRNAs

After the quantitative analysis was completed, an expression matrix of all samples was obtained, and edgeR software was used to analyze the expression differences. An lncRNA was considered differentially expressed if *p* < 0.05 and the log_2_(FC) was >2 or <−2. A volcano plot was constructed to illustrate the expression patterns of these lncRNAs in the acute cold stress and cold acclimation groups (Figure 4A,B). A total of 427 significant DElncRNAs were identified in the acute cold stress group (Appendix A), including 215 upregulated lncRNAs (50.4%) and 212 downregulated lncRNAs (49.6%). A total of 376 significant DElncRNAs were identified in the cold acclimation group (Appendix A), including 192 upregulated lncRNAs (51.1%) and 184 downregulated lncRNAs (48.9%). The acute cold stress-induced more lncRNAs. For the combined analysis, 113 lncRNAs were common to the two groups, with 58 upregulated lncRNAs and 55 downregulated lncRNAs (Figure 4C–E, Appendix A). The top 10 significant DElncRNAs of the two groups are listed in Table 1. TCONS_00000583 was significantly downregulated in the two groups.

### 3.4. Functional Prediction of DElncRNAs through Their Target Genes

Since lncRNAs do not encode proteins, their functions were predicted by their target genes. The target genes of DElncRNAs were initially identified. In the acute cold stress group, 427 DElncRNAs regulated 3994 genes, with 1422 genes (including 381 unannotated genes) in cis mode and 2572 genes (including 572 unannotated genes) in *trans* mode (Appendix A). In the cold acclimation group, 376 DElncRNAs regulated 3477 genes, with 1229 genes (including 320 unannotated genes) in *cis* mode and 2248 genes (including 507 unannotated genes) in *trans* mode (Appendix A). The regulation of target genes by lncRNAs in the two groups was nearly all in trans mode. There were many unannotated genes, indicating that the genome of Chinese pigs has not been completely annotated. The molecular mechanism of pig adaptation to low temperatures is complex and needs further research.

A functional enrichment analysis of the cis-regulated and trans-regulated genes with DElncRNAs was carried out. GO enrichment analysis was performed on the cis-regulated genes of DElncRNAs. The significantly enriched GO terms were defense response (GO:0006952), response to stress (GO:0006950), and cytokine receptor binding (GO:0005126) (Figure 5a). The trans-regulated mRNAs with DElncRNAs were enriched in the GO terms of immune system process (GO:0002376), transcription factor TFIID complex (GO:0005669), and nucleoside-triphosphatase regulator activity (GO:0060589) (Figure 5b). The KEGG pathway analysis indicated that the cis-regulated mRNAs were involved in 41 pathways (*q* < 0.05) (Appendix A). Signal transduction and the immune system were significantly affected. The trans-regulated mRNAs were involved in 117 pathways (*q* < 0.05) (Appendix A). Cell growth and death, signal transduction, and the immune system were significantly enriched. The top 20 pathways of cis-regulated mRNAs and trans-regulated mRNAs with DElncRNAs in the acute cold stress group are shown in Figure 5c,d, respectively.

Under cold acclimation, the GO analysis indicated that the most enriched GO terms targeted by the cis-regulated mRNAs were homophilic cell adhesion (GO:0007156), extracellular region (GO:0005576), and cytokine receptor binding (GO:0005126) (Figure 6a). The most enriched GO terms targeted by the trans-regulated mRNAs were immune response (GO:0006955) and cytokine activity (GO:0005125) (Figure 6b). The KEGG pathway analysis indicated that the cis-regulated mRNAs were involved in 18 pathways (*q* < 0.05) (Appendix A). Signal transduction and the immune system were significantly affected. The trans-regulated mRNAs were involved in 121 pathways (*q* < 0.05) (Appendix A). Cell growth and death, signal transduction, amino acid and carbohydrate metabolism, and the immune system were significantly affected. The top 20 pathways of cis-regulated mRNA and trans-regulated mRNA with DElncRNAs in the cold acclimation group are shown in Figure 6c,d, respectively.

### 3.5. Construction of an lncRNA-mRNA Network for the Top 10 Upregulated DElncRNAs

The lncRNA-mRNA networks for the top 10 upregulated DElncRNAs of the acute cold stress and cold acclimation groups were constructed using Cytoscape (Figure 7 and Figure 8). In the acute cold stress group, 10 DElncRNAs targeted 80 genes, and six cluster modules were obtained. In the cold acclimation group, 10 DElncRNAs targeted 48 genes, and five cluster modules were obtained. The functions of these DElncRNAs were primarily upregulated target genes. TCONS_00096631 was highly expressed in both groups. This induced the expression of the *TSPAN4* gene and suppressed the expression of *APOO*, *TDGF1*, and *ALS2CL* genes.

## 4. Discussion

Different animals have varied optimal temperatures, and there is variation within the same animal depending on age, weight, and stage of production. Each homeothermic animal has a suitable maximum critical temperature and a minimum critical temperature. If the temperature exceeds the critical temperature, the animal must begin to thermoregulate to maintain normal body temperature. At this time, regardless of temperature, physiological processes must respond to maintain homeostasis. With the rapid development of high-throughput sequencing technology, the cold resistance characteristics of many species have now been revealed at the genome and transcriptome levels [18,19,20].

LncRNAs are a type of non-coding RNA with lengths greater than 200 nt. LncRNAs are involved in a variety of biological processes, including DNA methylation, histone modification, RNA post-transcriptional regulation, and protein translation regulation, as well as in various physiological and pathological processes. They can act as cis-regulatory or trans-regulatory elements, meaning that they can not only regulate gene expression near their own transcription sites (cis-regulatory) but can also regulate gene expression at sites far away (trans-regulatory) [21]. Some lncRNAs have been found to play important roles in thermoregulation [22,23], but there are few such reports for pigs. This study used HiSeq 2500 high-throughput sequencing technology to analyze the role of lncRNAs in cis- and trans-regulatory responses to cold stress in pigs.

It has been demonstrated that low temperature alters the expression of lncRNAs and mRNAs in plants and animals [24,25,26,27]. When Enshi black pigs were exposed to low temperatures, acute cold stress-induced more DElncRNAs and response pathways compared to cold acclimation. However, the main energy substances employed to resist cold in both groups were the same, being sugars and amino acids. Acute cold exposure increases the energy utilization rate of peripheral tissues as well as glycolysis and gluconeogenesis of the liver, thus meeting the significantly increased energy demand and contributing to blood glucose homeostasis which is crucial for ensuring the energy supply of tissues [28]. Under acute cold stress, cold-tolerant Tibetan pigs use fatty acids, while cold-sensitive Bama pigs use glucose as the primary fuel source to protect against the cold [29]. Long-term cold acclimation usually suppresses the antioxidant defense system [30], slowing metabolism and delaying immunity [31]. Such acclimation generates higher quantities of triacylglycerols and fewer amounts of structural lipids and sphingomyelin, and the quantities of saturated and unsaturated fatty acids are altered [32]. Cold acclimation improves the efficiency of muscle oxidative phosphorylation and decreases proton leak and shivering intensity [33]. Cold acclimation can even alter the structure and composition of the heart muscle, but rewarming can reverse the process [34].

Under both acute cold stress and cold acclimation, DElncRNAs regulated target genes in a *trans* manner (Appendix A). The trans-regulated DEGs were significantly enriched in signal transduction, metabolism, and the immune system. In signal transduction, the Calcium signaling pathway, the HIF-1 signaling pathway, the Jak-STAT signaling pathway, the MAPK signaling pathway, the NF-kappa B signaling pathway, the Phospholipase D signaling pathway, the PI3K-Akt signaling pathway, the Sphingolipid signaling pathway, and the TNF signaling pathway were identified in both groups. To avoid physical damage or metabolic disruption under extreme temperatures, animals have evolved a set of sensory ion channels. Through alterations in channel conformation, ion flow undergoes changes, resulting in electrical signals and sensations [35]. The mechanism of signal transduction in protecting plants from low-temperature stress is relatively clear [36,37], but there is still limited research on this mechanism in animals.

Low temperature-induced cell membrane solidification may be sensed by proteins located on the cell membrane, and this rapidly induces an increase in Ca^2+^ concentration in the cytoplasm and generates specific Ca^2+^ signals. Low temperature-specific Ca^2+^ signaling is an important regulatory factor for the expression of low-temperature responsive genes [38]. Low temperatures can cause significant oxidative stress [39,40,41], and HIF-1 is a crucial oxygen-sensitive regulator of gene expression, not only to anoxia but also to freezing [42]. In mammals, HIF-1 plays important roles in glucose transport, glycolysis, and iron transport. Low temperature induces the expression of HIF-1 in fishes, insects, and rats [43,44,45]. In addition, the Hippo signaling pathway and the VEGF signaling pathway were affected in the acute cold stress group. The Hippo pathway was initially considered a master regulator of organ growth. However, the latest research suggests that Hippo signaling does not regulate normal growth [46]. This pathway was activated under acute cold stress, presumably related to its involvement in biological processes such as energy metabolism and oxidative stress [47]. VEGF primarily regulates angiogenesis. A hypoxic environment has implications for angiogenesis by stimulating vessel growth through the HIF-1α/VEGF axis [48]. The Rap1 signaling pathway and the Ras signaling pathway were enriched in the cold acclimation group. The enrichment of these two pathways implies changes in cell proliferation, survival, and differentiation [49].

Low temperatures can definitely cause changes in energy metabolism. Our results suggest that low temperature altered the metabolism of arginine, cysteine, methionine, tryptophan, and proline. The amino acid metabolism in rat BAT could be enhanced by cold temperature acclimation, and the metabolism and composition were more intensely altered after 15 days than after 12 h of cold exposure [50]. Long-term exposure to low temperatures leads to the metabolism of amino acids that increase structural flexibility and protein function [51]. This indicates that the small molecule system with amino acids is the main regulator of low-temperature stress [52]. In addition to amino acids, low temperature also affects the metabolism of carbohydrates such as starch, sucrose, and pyruvate. A short-term cold treatment altered the carbohydrate metabolism in the leaves of potatoes, leading to an increase in the content of soluble sugars [53]. Cold acclimation promotes starch hydrolysis while exhibiting higher sucrose levels [54]. Carbohydrates can directly influence cell membrane stability by interacting with the membrane interface and thereby support the maintenance of membrane integrity under freezing conditions [55].

Low temperatures can alter muscle energy metabolism, muscle fiber types, and immune factors. Studying the response mechanism of pig muscle tissue to low-temperature environments not only helps improve the quality of pork and cultivate cold-resistant pig varieties but also provides a theoretical foundation for the treatment of human diseases and for understanding human bodily responses to low-temperature conditions.

## 5. Conclusions

This study investigated the lncRNAs in pig muscle associated with low temperatures and constructed DElncRNA profiles. Low temperatures significantly altered the expression patterns of genes and lncRNAs in skeletal muscle. The DElncRNA and DEgene profiles of pig muscle under cold-acclimation conditions and under acute cold stress were different. However, the target genes were all primarily trans-regulated by DElncRNAs. According to functional analysis of the target genes, low temperatures changed the biological pathways of ion migration and the metabolism of amino acids and carbohydrates.

## Figures and Tables

**Figure 1 genes-14-01901-f001:**
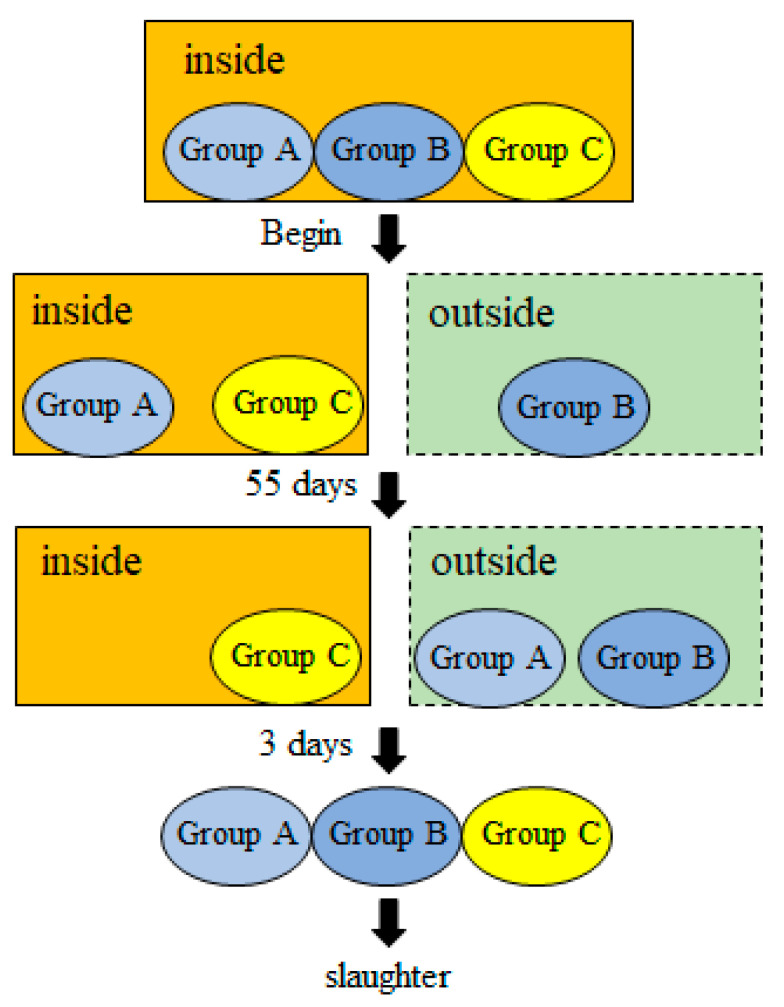
Experimental design scheme.

**Figure 2 genes-14-01901-f002:**
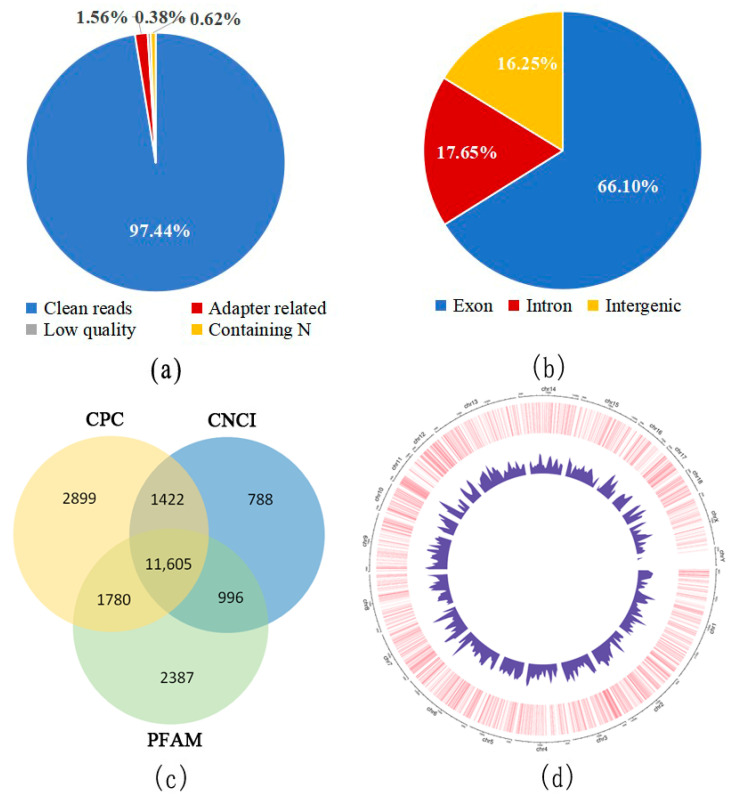
Identification of lncRNA in the dorsal muscle of Enshi black pigs. (**a**) Raw data composition. (**b**) Mapping region of clean reads. (**c**) Venn diagram for the non-coding RNAs. (**d**) Distribution of novel lncRNAs on each chromosome. The outermost circle represents the sizes of the pig chromosomes. The second circle indicates the location of the novel lncRNA on each chromosome. The smallest circle represents the density of novel lncRNAs on chromosomes.

**Figure 3 genes-14-01901-f003:**
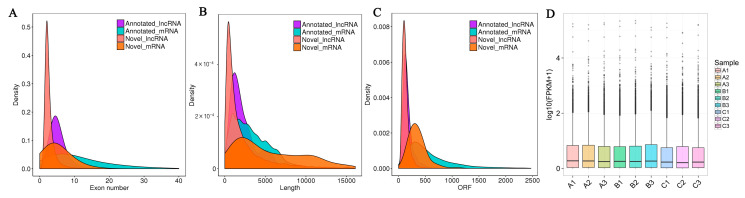
Specific characteristics of Enshi black pig lncRNAs. (**A**) Enshi black pig lncRNAs and mRNA exon number. (**B**) Enshi black pig lncRNAs and mRNA length distribution. (**C**) Enshi black pig lncRNAs and mRNA ORF length. (**D**) Box plot of expression levels of different samples. The abscissa is the sample name, and the ordinate is log10 (FPKM+1). The box plot of each area corresponds to five statistics (from top to bottom, these are the maximum value, the upper quartile, the median value, the lower quartile, and the minimum value).

**Figure 4 genes-14-01901-f004:**
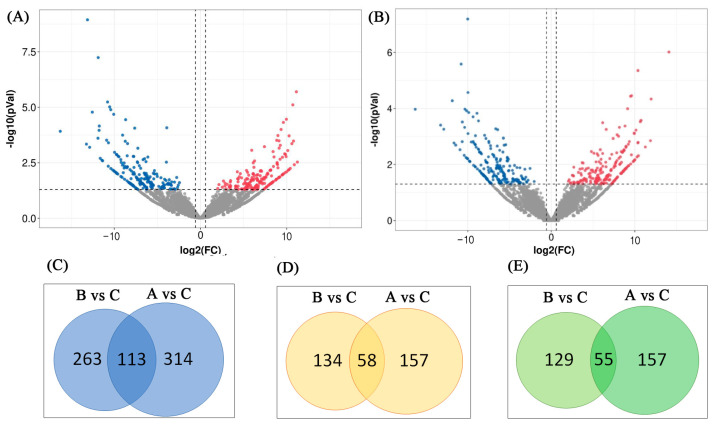
DElncRNAs in the dorsal muscle of Enshi black pigs after cold treatment. (**A**) Volcano plot for DElncRNAs in the acute cold stress group. Red indicates significantly upregulated lncRNAs (*p* < 0.05); blue indicates significantly downregulated lncRNAs (*p* < 0.05), and gray indicates lncRNAs for which the differences were not statistically significant (*p* > 0.05). (**B**) Volcano plot for DElncRNAs in the cold acclimation group. The color markings have the same meanings as in (**A**). (**C**) Common DElncRNAs (|log_2_FC| > 2 and *p* < 0.05) in the cold acclimation group and acute cold stress group. (**D**) Upregulated DElncRNAs in the cold acclimation group and acute cold stress group. (**E**). Downregulated DElncRNAs (|log_2_FC| > 2 and *p* < 0.05) in the cold acclimation group and acute cold stress group.

**Figure 5 genes-14-01901-f005:**
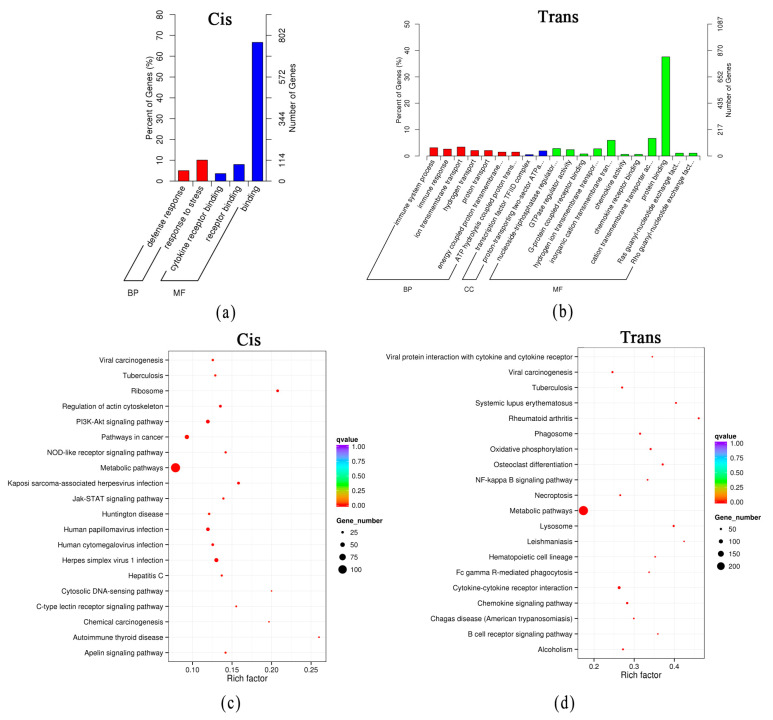
Functional prediction of DElncRNAs in the acute cold stress group by target genes. (**a**) GO analysis of cis mRNAs with DElncRNAs in the acute cold stress group. (**b**) GO analysis of trans mRNAs with DElncRNAs in the acute cold stress group. (**c**) The top 20 pathways of cis mRNAs with DElncRNAs in the acute cold stress group. (**d**) The top 20 pathways of trans mRNAs with DElncRNAs in the acute cold stress group.

**Figure 6 genes-14-01901-f006:**
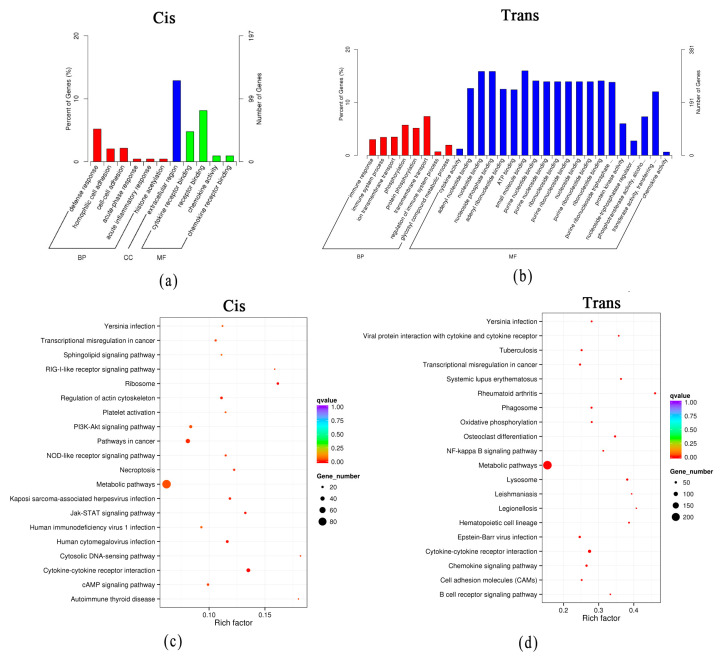
Functional prediction of DElncRNAs in the cold acclimation group by target genes. (**a**) GO analysis of cis mRNAs with DElncRNAs in the cold acclimation group. (**b**) GO analysis of trans mRNAs with DElncRNAs in the cold acclimation group. (**c**) The top 20 pathways of cis mRNAs with DElncRNAs in the cold acclimation group. (**d**) The top 20 pathways of trans mRNAs with DElncRNAs in the cold acclimation group.

**Figure 7 genes-14-01901-f007:**
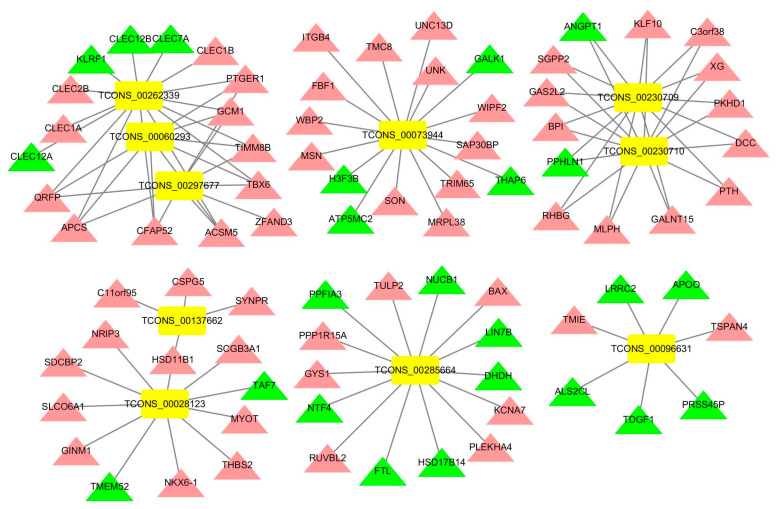
lncRNA-mRNA network for the top 10 upregulated DElncRNAs of the acute cold stress group. Rectangles represent lncRNAs, triangles represent genes, red represents up-regulated genes, green represents down-regulated genes, and yellow represents up-regulated lncRNAs.

**Figure 8 genes-14-01901-f008:**
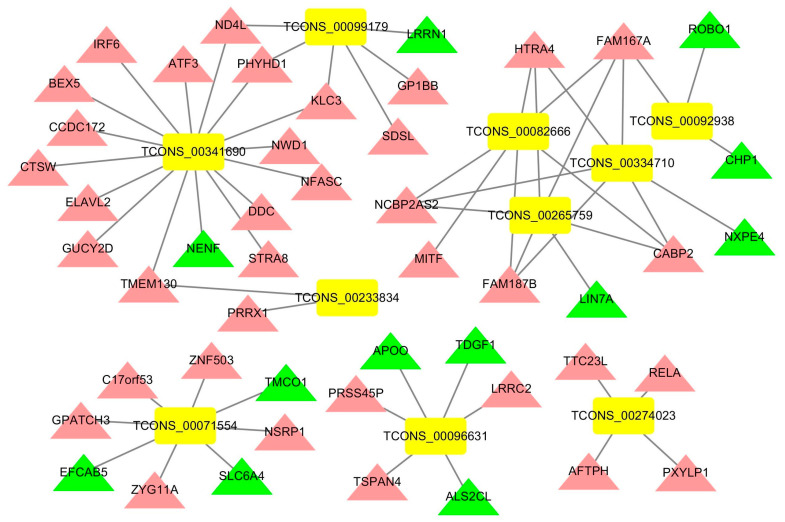
lncRNA-mRNA network for the top 10 upregulated DElncRNAs of the cold acclimation group. Rectangles represent lncRNAs, triangles represent genes, red represents up-regulated genes, green represents down-regulated genes, and yellow represents up-regulated lncRNAs.

**Table 1 genes-14-01901-t001:** DElncRNAs in Enshi black pig dorsal muscle after cold treatment.

A vs. C Group				B vs. C Group			
lncRNA	FDR	|log_2_FC|	Regulated	lncRNA	FDR	|log_2_FC|	Regulated
Linc1620	0.0006	14.09	up	Linc862	2.83 × 10^−6^	13.11	down
Linc361	0.0118	11.96	up	TCONS_00340242	0.0065	12.55	down
TCONS_00349979	0.0127	11.87	down	TCONS_00344988	0.0001	11.86	down
TCONS_00000583	0.0014	10.80	down	TCONS_00233834	0.0015	11.12	up
TCONS_00073944	0.0021	10.38	up	TCONS_00000583	0.0031	10.78	down
Linc1918	0.0001	10.01	down	Linc1207	0.0037	10.71	up
Linc3782	0.0083	9.98	down	Linc938	0.0042	10.53	down
TCONS_00284293	0.0098	9.62	up	Linc2552	0.0053	10.41	down
TCONS_00244588	0.0102	9.49	up	TCONS_00275432	0.0080	10.07	down
TCONS_00050025	0.0194	9.14	up	Linc1680	0.0123	9.96	up

Note: FDR, false discovery rate.

## Data Availability

The original contributions presented in the study are included in the article/Supplementary Material; further inquiries can be directed to the corresponding author.

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
