# Peer review of "The Role of lncRNAs in Pig Muscle in Response to Cold Exposure"

_genes, 2023, doi:10.3390/genes14101901_

Round 1
Reviewer 1 Report
The paper deals with the lncRNAs in pig. The topic is hot, and the article is well made. Some minor corrections are needed.
The control and experimental groups had only 3 animals, emphasize it at the end of the manuscript.
The authors refer to the supplemental material, but I did not it at disposal. Why?
R. 56, gene abbreviation in italic.
Fig. 1, check the format it is unreadable.
Have you analysed also the interactions of lncRNAs – siRNAs, miRNAs? Mention it in the R or D section.
Figs. 2,3, check the format, they are unreadable. The same and more for Fig. 5.
Explain the abbreviation lincRNA, r. 145. There is some difference between lnc – lincRNA.
Table 1, explain FDR.
Arrange better the Figures 7,8. Add legend, explain colors, shapes.
Author Response
Q: The control and experimental groups had only 3 animals, emphasize it at the end of the manuscript.
A: Each group included three individuals, which are introduced in the materials and methods section. (line 82)
Q: The authors refer to the supplemental material, but I did not it at disposal. Why?
A: Supplemental material are included in the “manuscript.zip” file.
Q: R. 56, gene abbreviation in italic.
A: I had read through the entire text and changed the gene names to italic.
Q: Fig. 1, check the format it is unreadable.
A: I had changed the format of Fig.1.
Q: Have you analysed also the interactions of lncRNAs – siRNAs, miRNAs? Mention it in the R or D section.
A: Sorry, I didn’t sequence the siRNA and miRNA, only sequenced the lncRNA and mRNA.
Q: Figs. 2,3, check the format, they are unreadable. The same and more for Fig. 5.
A: I had changed the format of Fig.2, Fig.3 and Fig.5.
Q: Explain the abbreviation lincRNA, r. 145. There is some difference between lnc – lincRNA.
A: Yes, I had explained the abbreviation lincRNA in line 145.
Q: Table 1, explain FDR.
A: Yes, I had explained the FDR in Table 1 note.
Q:Arrange better the Figures 7,8. Add legend, explain colors, shapes.
A:Yes, I had added legend in Fig.7,8.
Reviewer 2 Report
Congratulations!
The study is on pigs muscle, as tissue for produce heat through shivering thermogenesis and non-shivering thermogenesis under cold exposure and on long non-coding RNAs (lncRNAs) wich play important roles in regulating gene expression.
A total of 11605 non-coding RNAs were obtained, including 10802 novel lncRNAs. 113 lncRNAs were commonly upregulated by acute cold and cold acclimation and 65% of the target genes were trans-regulated by DElncRNAs.
Compared to cold acclimation, acute cold stress induced more DElncRNAs and response pathways. In conclusion, low temperature altered the expression levels of lncRNAs and their target genes in muscle tissue. Some potential mechanisms were revealed, including ion migration and the metabolism of amino acids and carbohydrates.
Suggestions:
Line 42-43: The Mangalica pig, the only pig breed with long hair, 42 lives in the colder regions of Hungary [3] (and much better Romania where is colder than Hungary!
Figure 2.D. have to be enlarged or it can became Figure 3. much larger.
Author Response
Q: Line 42-43: The Mangalica pig, the only pig breeds with long hair, 42 lives in the colder regions of Hungary [3] (and much better Romania where is colder than Hungary!
A: I had added this sentence in line 42-43.
Q:Figure 2.D. have to be enlarged or it can become Figure 3. much larger.
A:Fig.2 has been reformatted.
Reviewer 3 Report
Major issues
Please describe the gene ontology procedures, with more details. As it is now, this section is not helping readers to understand fully the results of the study.
Minor issues
The introduction deals with three different ideas, so I suggest to separate into three different sub-sections.
The objectives of the work must be presented clearly.
All the figures in the manuscript must be replaced with new ones that be read easily (small size, small letters etc.)
In discussion, please add a passage with the clinical importance of the findings.
Please add a concluding section.
Author Response
Q: Please describe the gene ontology procedures, with more details. As it is now, this section is not helping readers to understand fully the results of the study.
A: Yes, I had described the gene ontology procedures in line 137-145.
Minor issues
Q: The introduction deals with three different ideas, so I suggest to separate into three different sub-sections.
A: Yes, I had separate them into three different sub-sections. (line49-62)
Q: The objectives of the work must be presented clearly.
A: The objectives of the work had been added at the end of introduction. (line 79-83)
Q: All the figures in the manuscript must be replaced with new ones that be read easily (small size, small letters etc.)
A: Yes, I had reformatted all the figures.
Q: In discussion, please add a passage with the clinical importance of the findings.
A:I added a passage with the clinical importance of the findings.(line 370-374)
Q:Please add a concluding section.
A: I added a concluding section at the end of manuscript.
Round 2
Reviewer 3 Report
The manuscript has been improved.
The authors need to correct formatting before final approval. Also, extensive revision of language needs to take place.
The manuscript has significant linguistic flaws. I understand that it is difficult to write perfectly in a foreign language and I am supportive in this regard,
but please do look into grammar and wording of passages throughout the text and please do make all necessary improvements for the flow of the reading of the final manuscript.
Author Response
I had correct formatting and revised English.